# Learning Non-Local Phonological Alternations via Automatic Creation of Tiers

## Abstract

Phonological alternations often involve dependencies between adjacent segments. Despite the apparent long-distance nature of alternations such as consonant and vowel harmony, even these can be reduced to dependencies between adjacent segments by projecting a subset of segments onto a new representation, often called a tier. Tiers are known to simplify learning non-local dependencies and are consistent with human behavior in artificial language experiments. However, little is known about the mechanism by which learners may construct such a representation. In this work, we propose a computational model that learns non-local alternations by automatically constructing alternation-relevant tiers. The model is sensitive to adjacent dependencies and—when adjacency fails—uses this same sensitivity to construct a tier that reduces the relevant non-local dependencies to local ones, falling within its purview. The model accurately matches the behavior of humans on prior artificial language experiments. This submission describes preliminary work and is intended as an extended abstract.

## 1 Introduction

Phonological segments often alternate in a way that is predictable from phonological environment. For instance, the English plural (Pʟ) affix alternates between [-z] and [-s], matching the voicing of the final segment of the Sɢ form, as exemplified in (1).

(1)  /dɑg-Pʟ/ → [dɑgz]
     /kæt-Pʟ/ → [kæts]

Such *phonological alternations* are common across the world's languages, and often involve dependencies between adjacent segments.

However, consonant harmony (Rose and Walker, 2004) and vowel harmony (Van der Hulst, 2016) are phonological alternations that often involve dependencies between segments that are arbitrarily far away. For instance, Turkish affix vowels

Figure 1: The proposed model tracks only adjacent dependencies. Any adjacent dependencies that fail are deleted until a sufficiently accurate generalization can be formed in terms of adjacent segments § 2.

match the feature [±back] to the final stem vowel, as shown in (2) (examples from Kabak, 2011, p. 3 and Nevins, 2010, p. 28); the affix vowels alternate between back {ɑ, ɯ} and front {e, i} to match the [±back] value of the preceeding vowel. Arbitrary numbers of consonants can intervene.

(2)  [dɑl-lɑr-ɯn]    branch-Pʟ-Gᴇɴ
     [jer-ler-in]    place-Pʟ-Gᴇɴ
     [ip-ler-in]     rope-Pʟ-Gᴇɴ

The Omotic language Aari exhibits a sibilant harmony pattern exemplified in (3) from McMullin (2016, p. 21) (adapted from Hayward 1990). Underlying /s/ (3a) surfaces as [ʃ] when it is preceded by a [−ant] sibilant at any distance (3b) (dependent sibilants are underlined for clarity).

(3)  a.  /baʔ-s-e/    → [baʔse] 'he brought'
     b.  /ʔuʃ-s-it/   → [ʔuʃʃit] 'I cooked'
         /ʒaʔ-s-it/   → [ʒaʔʃit] 'I arrived'
         /ʃed-er-s-it/ → [ʃederʃit] 'I was seen'

Experiments on sequence learning have forcefully demonstrated that learners more readily track dependencies between adjacent segments (Saffran et al., 1996, 1997; Aslin et al., 1998) than non-adjacent segments (Santelmann and Jusczyk, 1998; Gómez, 2002; Newport and Aslin, 2004; Gómez and Maye, 2005). Such results suggest that learning non-local alternations, such as those in (2)-(3)

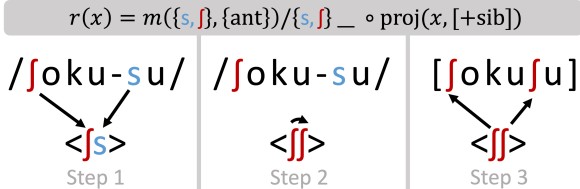

$r(x) = m(\{s, \int\}, \{ant\})/\{s, \int\} \_ \circ proj(x, [+sib])$

| /ʃoku-su/ | /ʃoku-su/ | [ʃokuʃu] |
|---|---|---|
| <ʃs> | <ʃʃ> | <ʃʃ> |
| Step 1 | Step 2 | Step 3 |

Figure 2: A rule generates an output sequence in three steps: (1) it projects a tier, (2) it applies feature matching over that projection, and (3) it replaces original segments with their results from the tier projection.

is more challenging than learning local alternations like (1). However, since at least Goldsmith (1976), researchers have recognized that non-local alternations like (2)-(3) can be reduced to local alternations by projecting a subset of segments onto a relevant phonological tier.

Moreover, formal-language-theoretic analysis (Heinz et al., 2011; McMullin, 2016; Jardine and Heinz, 2016; Jardine and McMullin, 2017) and computational modeling experiments (Hayes and Wilson, 2008) have provided evidence that phonological tier is a representation that benefits learning, and artificial language experiments have provided evidence that learners may indeed construct tiers (Finley, 2011; McMullin and Hansson, 2019). However, the tier needed to render dependencies local varies heavily depending on what alternation is being learned (Nevins, 2010; Burness et al., 2021), and the mechanism by which learners may construct tiers remains largely unknown.

In this work, we propose a computational model § 2 that automatically constructs a tier and discovers alternations by tracking only adjacent dependencies (Fig. 1). The model learns tiers consistent with human behavior in artificial language experiments and learns non-local alternations in preliminary natural language settings § 3.

## 2 Model

Our model is called D2L for *Distant To Local* because it learns non-local alternations by automatically constructing a tier that exposes distant dependencies as local. D2L takes as input a set $V$ of $(x, y)$ input-output pairs, along with a set $A$ of alternating segments and what features $F$ of those segments alternate. D2L tracks segments adjacent to the alternating segments and attempts to generalize from these which alternating segment of $A$ surfaces. If adjacent segments fail to account for the alternation, D2L deletes the adjacent segments that do

not work, by projecting the other segments onto a new tier. This processes repeats iteratively until a successful generalization is found. D2L's generalizations are implemented as rules, which produce an output sequence from an input sequence.

### 2.1 Rule Structure

An example[1] sibilant harmony rule is exemplified in Fig. 2, where it projects a sibilant tier (step 1), harmonizes the tier-adjacent sibilants (step 2), and outputs an updated sequence (step 3).

We characterize a rule $r(x) = r_{adj} \circ proj(x, T)$ as a local rule $r_{adj}$ applied over a tier projection, $proj(x, T)$. The local rule can either match the features of a target $A$ to a left neighbor $r_{adj} = m(A, F)/L\_\_$ or a right neighbor $r_{adj} = m(A, F)/\_\_R$ on the tier projection. We treat segments as bundles of distinctive features, where each feature $f$ has a value $\alpha$ (e.g., '+', '−'). The feature-matching operation $m(A, F)$ sets the $\alpha$ value of any segment in $A$ to match those of a segment in $L/R$ for each feature $f \in F$; e.g., $m(\int, \{ant\})/s\_\_ = s$.

The rule applies iteratively to allow for modeling the spreading patterns that are pervasive in vowel and consonant harmony (Nevins, 2010; Burness et al., 2021). If the rule has a left context, it applies left-to-right; otherwise right-to-left.[2]

Tier projection is inspired by the concept of an *erasing function* (Heinz et al., 2011) (4), projecting a length-$n$ sequence $x$ to a new, order-preserving sequence that preserves only the $x_i \in T$.

(4)  $proj(x, T) \triangleq t_1, ..., t_m : x_i \in t$ iff $x_i \in T$

In Fig. 2, $proj(/\int okusu/, [+sib]) = <\int s>$. We extend this with a set of links, E(x, t), from input segments to their tier projections (if they have one). In Fig. 2, $E(x, t) = \{(x_1, t_1), (x_5, t_2)\}$. The final output sequence is generated by iterating over each $(x_i, t_j) \in E(x, t)$ and replacing each $x_i$ with $t_j$.

### 2.2 Learning

We give D2L's pesudocode in Alg. 1, and visualize a toy example in Fig. 3. The input is a set of input-output pairs, and a set of alternating segments $A$ and their alternating features $F$ (which are any features that vary across the segments in $A$). D2L first initializes the tier to contain all segments. Consequently, local generalizations like (1) are a special

---

[1]Examples in Figures 2-3 use artificial language data from Finley (2011); cf. § 3.1.1 for details.

[2]Spreading in both directions can be modeled with two rules, one applied in each direction.

**Algorithm 1** D2L

**Input**: pairs $V$, alternating segs $A$ and feats $F$
1: $T \leftarrow \Sigma$            ▷ Initialize tier to all segments
2: $D \leftarrow \emptyset$           ▷ Initially delete no segments
3: **while** $T \neq \emptyset$ **do**
4:     $L \leftarrow \{s : s \text{ precedes some } a \in A \text{ on } T\}$
5:     $R \leftarrow \{s : s \text{ follows some } a \in A \text{ on } T\}$
6:     $r_l = m(A, F)/L\_\_ \circ \text{proj}(\cdot, T)$    ▷ Left rule
7:     $r_r = m(A, F)/\_\_R \circ \text{proj}(\cdot, T)$   ▷ Right rule
8:     $r = \arg\max_{r \in \{r_l, r_r\}} \text{acc}(r)$
9:     **if** $\text{acc}(r) > \theta$ **then**
10:       **return** $r$
11:     $N \leftarrow L \cup R$
12:     $D \leftarrow D \cup \{s \in N : \text{harmonizing with } s \text{ fails}\}$
13:     $C \leftarrow \arg\min_{\{\text{nat class } C : D \subseteq C \wedge A \cap C = \emptyset\}} |C|$
14:     $T \leftarrow T \setminus C$

---

**INPUT**

$V \begin{cases} \text{/ʃupe-su/} \rightarrow \text{[ʃupeʃu]} \\ \text{/ʃoku-su/} \rightarrow \text{[ʃokuʃu]} \\ \text{/ʃito-su/} \rightarrow \text{[ʃitoʃu]} \end{cases}$    $A = \{s, ʃ\}$ 
 $F = \{\text{ant}\}$

$T = \Sigma$    <ʃupesu> <ʃokusu> <ʃitosu>

*Iteration 1*    $N = \{e, u, o\}$     $D = \{e, u, o\}$

$T = [+\text{cons}]$    <ʃps>   <ʃks>   <ʃts>

*Iteration 2*    $N = \{p, k, t\}$   $D = \{e, u, o, p, t\}$

$T = [+\text{sib}]$    <ʃs>   <ʃs>   <ʃs>

*Iteration 3*

---

**Output**: $r(x) = m(\{s, ʃ\}, \{\text{ant}\})/\{ʃ\} \_\_ \circ \text{proj}(x, [+\text{sib}])$

Figure 3: Example of D2L.

case of D2L. Next, the while loop begins. Left and right rules are constructed with the current tier $T$; the most accurate is selected and compared to a threshold $\theta$. This threshold could be 100% accuracy or something more tolerant to exceptions, like the Tolerance Principle (Yang, 2016). If the rule is accurate enough, it is returned and the algorithm is finished. Otherwise the tier is updated § 2.2.1.

#### 2.2.1 Updating the Tier

Because the segments, $A$, must be targeted by the rule, they must be on the tier. The set $N$ contains all segments adjacent to an alternating segment on the current tier projection. A subset of these, $D \subseteq N$, contains any tier-adjacent segment that cannot be harmonized with. This can be because the segment is not specified for the feature(s) $F$ (e.g., a sibilant

cannot take an 'ant' feature from a vowel), or because it leads to the wrong surface segment (e.g., if /s/ takes /p/'s 'ant' feature, it will surface as [s], but may have been supposed to surface as [ʃ]). These segments cannot be on the tier. Thus, D2L takes the smallest natural class $C$ that contains all of $D$ but none of $A$, and removes this from the tier.

## 3 Experiments

### 3.1 Human-Like Behavior

We first compare D2L to human learners in Finley (2011)'s artificial language experiment.

#### 3.1.1 Background

Finley (2011) presented participants with training data: a <stem, suffixed> pair where the suffix sibilant harmonized with the stem sibilant across a single intervening vowel (5a). Learners would not generalize to novel cases where non-sibilant consonants also intervened (5b), choosing—in a forced-choice two-alternative test—the non-harmonizing option as often as a control group did.

(5)    a.   /diso-su/ → [disosu]
          /nesi-su/ → [nesisu]
          /piʃa-su/ → [piʃaʃu]
          /kuʃo-su/ → [kuʃoʃu]
   b.   /ʃuko-su/ → [ʃukosu]

These results are consistent with the view that learners constructed a [+cons] tier, since this would render the training sibilants (5a) adjacent, but harmony for the test sibilants (5b) would be blocked by the non-sibilant consonants.

In contrast, if presented with training data where sibilants harmonize across both intervening vowels and non-sibilant consonants (6a), learners generalized to cases where only a vowel intervened (6b).

(6)    a.   /suge-su/ → [sugesu]
          /sone-su/ → [sonesu]
          /ʃupe-su/ → [ʃupeʃu]
          /ʃako-su/ → [ʃakoʃu]
   b.   /kuʃa-su/ → [kuʃaʃu]

In this case, the results are consistent with the view that learners constructed a [+sib] tier, since this is needed to render the training sibilants (6a) adjacent, and the test sibilants (6b) would also be adjacent on the [+sib] tier. Together, these results suggest that learners construct tiers in response to the data they are exposed to.

### 3.1.2 Setup

We ran D2L on the 25 train instances from Finley (2011), for each setting (5)-(6) described above (§ 3.1.1). We treated the underlying affix sibilant as unspecified [±ant], and used D2L's output as its choice in the forced-choice two-alternative test.

As comparison models, we used Goldsmith and Riggle (2012), which we call GR, and the finite-state model of Jardine (2016); Jardine and Heinz (2016), which we call FS. GR is an information-theoretic model, which uses a two-state hidden markov model to induce a tier, and then fits a probabilistic phonotactic model with a Boltzmann distribution, which incorporates information from both string and tier bigrams. FS attempts to induce a tier strictly-local finite-state acceptor (FSA) that can accept or reject strings as grammatical. We used the python implementation from (Aksënova, 2020). Since these models are phonotactic models, we ran both choices of the forced-choice two-alternative test through the models and used the one that is scored higher as the models' respective choices.

### 3.1.3 Results

Results are reported in Tab. 1. The HUM row gives human results, with a '✓' wherever the experimental group chose the harmonizing test choice over the non-harmonizing choice significantly more often than the control group (as measured in the original study); '✗' appears elsewhere. For computational models, '✓' marks results where the model chose the harmonizing test choice over the non-harmonizing at a rate significantly greater than chance ($\approx > 50\%$ accuracy).

Only D2L matches the results of human learners, learning a [+cons] tier in the first experiment condition and a [+sib] tier in the second.

Neither GR nor FS succeed in learning a meaningful tier on 25 instances, and consequently cannot generalize from the training data to test instances—even of the type that humans did.

McMullin and Hansson (2019) performed similar experiments, involving regressive liquid harmony. D2L is the only model to match human performance on those experiments as well, but we omit the results due to space limitations.

### 3.2 Learning Natural Language Alternations

We present preliminary results running D2L on two natural language alternations: vowel harmony in Turkish (Kabak, 2011) and Finnish (Ringen and Heinämäki, 1999). Both languages exhibit a

Table 1: Finley (2011) Sibilant Harmony

| | Train CVS̲V-S̲V | | Train S̲VCV-S̲V | |
|---|---|---|---|---|
| Model | CVS̲V-S̲V | S̲VCV-S̲V | S̲VCV-S̲V | CVS̲V-S̲V |
| HUM | ✓ | ✗ | ✓ | ✓ |
| D2L | ✓(1.0) | ✗(0.0) | ✓(1.0) | ✓(1.0) |
| GR | ✗ | ✗ | ✗ | ✗ |
| FS | ✗ | ✗ | ✗ | ✗ |

Table 2: Turkish and Finnish Vowel Harmony

| Language | Tier Learned | Accuracy |
|---|---|---|
| Turkish | [−cons] | 1.0 |
| Finnish | [−cons] \ {i, e} | 1.0 |

[±back] vowel alternation: affix vowels harmonize with the closest stem vowel. All vowels participate in Turkish; {i,e} are neutral in Finnish. Using frequency-annotated data from the MorphoChallenge (Kurimo et al., 2010), we ran D2L on the 200 most frequent words from each language (separately). As a proxy for morphological boundaries, we treated each word's first vowel as fully specified and treated all following vowels as underspecified for their [±back] feature (Van der Hulst, 2016).[3] We treat underlyingly unspecified vowels as always harmonizing. These results are preliminary; future versions will improve on the realism of the setup.

We show the tier that D2L learned and the accuracy of the resulting rule in Tab. 2. Accuracy was computed over the 1K most frequent words, meaning the 200 training words and 800 novel words. D2L learns an appropriate tier for each language: the vowel tier for Turkish, and the vowel tier minus the neutral vowels {i,e} for Finnish.

## 4 Conclusion and Discussion

We presented a computational model, D2L, which tracks adjacent segments in order to account for phonological alternations. When these segments are inadequate, D2L deletes the adjacent segments that failed to account for the alternation and tracks the resulting, newly-adjacent dependencies. This process, carried out iteratively until an adequate generalization is discovered, learns non-local phonological alternations consistent with human behavior in artificial language experiments. Local alternations constitute a special case, where no tier is needed. Future work will apply D2L to a more diverse range of natural language settings.

---

[3]Turkish has secondary rounding harmony for high vowels, which we intend to model in future versions of this work.

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
