# OpenReview forum: "Learning Non-Local Phonological Alternations via Automatic Creation of Tiers"
_aclweb.org/ACL/2022/Workshop/CMCL — CMCL 2022 nonarchival_

### Official Review · Reviewer_tYg5 · 2022-03-22
**Learning Non-Local Phonological Alternations via Automatic Creation of Tiers**

**Rating:** 7
**Confidence:** 4

**Review:**

The article presents a computational model for dealing with phonological alternations. It presents a cognitive argument for why this is interesting, but could also use a computational argument for why this is interesting, i.e., how might this model be used? Overall it is very well written and easy to understand. The only problems I had with the article were about formatting:

1. Place Figures and Tables below the paragraphs that you first mention them in. For instance, Figure 1 is on the first page but not mentioned until the second, while Table 2 is before the paragraph starting on line 271.
2. As far as I know (although if I'm wrong please ignore this), the symbol, "§", is not required of you to use. I was quite confused about what it was supposed to mean and you don't use it consistently. For a while I thought you were trying to use a symbol for the word 'Figure'. After clicking on it I grasped that you're using it to mark the presence of a footnote, but then later you also use it to denote a section in the main text.
In terms of footnotes, seeing as the citations are spelled out, i.e., not superscripted, there's no reason to use a confusing symbol that takes up space. I recommend you use superscripts for footnotes and delete the "§" (as you did on line 267). That's my first opinion. My second opinion is that you can do without footnotes entirely. You don't say anything that can't be added to the main text or left out. For instance, footnote 1 is there just to tell you that in 3.1.1 there is more info... You don't need a footnote for that. I would even say that footnote 2 is something that should be in the main text as its an important detail.
In terms of using the "§" symbol to denote sections in the main text... I suppose the question would be "why?" Does it facilitate reading? On line 153, does it make reading easier to tell the reader they can find out more in the section that immediately follows? My first opinion is that you get rid of these as I don't understand what purpose they serve. My second opinion is that if you really want to keep them then you need to follow writing logic and put them in parentheticals because they do not belong within the syntax of the sentences you are placing them in. You did this correctly on line 211.

Below are some minor writing mistakes
1. line 028: 'phonological environments'
2. line 072: 'a phonological tier is a representation', or 'phonological tiers are representations'
3. line 102: 'process'
4. line 112, 'a local rule, 𝑟adj, applied'
5. line 228: 'the one that scored higher'
6. line 239: 'over the non-harmonizing choice'
7. line 246 and 247: 'and consequently could not generalize from the training data to test instances even when humans did'
8. line 258: 'All vowels that participate'
9. line 273 and 274: 'consisting of 200 training words and 800 novel words'
10. line 275: 'D2L learned'

---

### Official Review · Reviewer_pZXL · 2022-03-25
**Interesting but the model could benefit from more phonological knowledge?**

**Rating:** 6
**Confidence:** 1

**Review:**

This paper reports a model that learns distant phonological alternations. The results of the model are overall comparable to human learners' performance. I think the paper is quite interesting, but to be honest I'm not sure what would be the application of this model. In natural language, non-local alternations are often conditioned by the phonological context (e.g., assimilation, which is the case in the real-language examples cited in this paper). The current model doesn't seem to have phonological knowledge. I'm curious to know what would be the next step of this project. Is the goal to simulate human learners?

---

### Official Review · Reviewer_ztMY · 2022-03-25
**Learning non-local alternations via tier creation**

**Rating:** 7
**Confidence:** 5

**Review:**

The authors propose a model for learning non-local phonological alternations via the creation of feature tiers. The work sits at the intersection of tier-based analyses of phonotactic computational complexity (Heinz et al) and artifical grammar learning (AGL) experiments on learning of local and nonlocal phonological processes. The submission is as an extended abstract.

The authors provide pseudocode for the functioning of the model, walk the reader through an example of its functioning on some data, and discuss its relevance to human performance on previous AGL experiments. They also compare the model's performance against those of competing HMM and FSM based models.

The authors make a number of claims that require non-trivial support e.g. (i) that learning non-local is harder than local, which may be true in AGL scenarios but this says little or nothing about natural language acquisition scenarios, or (ii) claims from model learning results that human learners construct tiers.

There is a fair amount of unclarity in the formalism and notation: it feels like much of this could have been eliminated from this submission but would benefit a subsequent lengthier submission where there is more room for definitions and examples.

Some specific notes:
- opening discussion of English plural morph refers to it as a phonological segment (omitting discussion of the [-Id] realization)
- the composition of V should be clarified in line 94 (i.e. input-output pairs of what; this is unclear from context)
- if segments are treated as bundles of features (line 117) then it's not clear why segments continue to be referred to
- it's not clear how the final tier is written back to the final output form; the tier projection operation as described seems to be lossy (the "final sequence" in lines 135-137 appears to only be the final tier representation)
- the input V (set of input-output pairs) to Algorithm 1 appears to not be used/referenced anywhere in the algorithm
- the notion of "accuracy" referenced on line 148 and in Algorithm 1 is not clear; accuracy at what?
- the work is reminiscent of the search/copy approach (Samuels, Nevins, Reiss/Mailhot) to phonological operations (including the observation that strict locality is merely a special case of non-locality), only with unnecessary notational baggage and it's not clear what it's benefit is

Notwithstanding all of the above, the paper is a promising start to work that could eventually make contact with the tier-based formal work from the Stony Brook group providing some linking hypotheses across Marrian levels of phonology. It would benefit from being presented and critiqued and I'd like to see it accepted and presented.

---

### Decision · Program_Chairs · 2022-03-29

Accept (non-archival)